# A Quality Improvement Initiative to Reduce Bronchopulmonary Dysplasia in a Level 4 NICU—Golden Hour Management of Respiratory Distress Syndrome in Preterm Newborns

**DOI:** 10.3390/children8040301

**Published:** 2021-04-15

**Authors:** Andrew M. Dylag, Jamey Tulloch, Karen E. Paul, Jeffrey M. Meyers

**Affiliations:** 1Division of Neonatology, Department of Pediatrics, Golisano Children’s Hospital, University of Rochester Medical Center, 601 Elmwood Avenue, Box 651, Rochester, NY 14642, USA; jamey_tulloch@urmc.rochester.edu (J.T.); jeff_meyers@urmc.rochester.edu (J.M.M.); 2Department of Nursing, University of Rochester Medical Center, Rochester, NY 14642, USA; karen_paul@urmc.rochester.edu

**Keywords:** quality improvement, bronchopulmonary dysplasia, chronic lung disease

## Abstract

Background: Prevention of chronic lung disease (CLD) requires a multidisciplinary approach spanning from the delivery room to Neonatal Intensive Care Unit (NICU) discharge. In 2018, a quality improvement (QI) initiative commenced in a level 4 NICU with the goal of decreasing chronic lung disease rates below the Vermont Oxford Network (VON) average of 24%. Methods: Improvement strategies focused on addressing the primary drivers of ventilation strategies, surfactant administration, non-invasive ventilation, medication use, and nutrition/fluid management. The primary outcome was VON CLD, defined as need for mechanical ventilation and/or supplemental oxygen use at 36 weeks postmenstrual age. Statistical process control charts were used to display and analyze data over time. Results: The overall CLD rate decreased from 33.5 to 16.5% following several interventions, a 51% reduction that has been sustained for >18 months. Changes most attributable to this include implementation of the “golden hour” gestational age (GA) based delivery room protocol that encourages early surfactant administration and timely extubation. Fewer infants were intubated across all GA groups with the largest improvement among infants 26–27 weeks GA. Conclusions: Our efforts significantly decreased CLD through GA-based respiratory guidelines and a comprehensive, rigorous QI approach that can be applicable to other teams focused on improvement.

## 1. Introduction

Bronchopulmonary Dysplasia (BPD) and Chronic Lung Disease (CLD) are among the most common complications of premature birth. Healthcare costs related to extreme prematurity and chronic lung disease extend beyond the NICU hospitalization and increase with BPD [1,2]. Multiple factors contribute to BPD/CLD including mechanical ventilation, oxygen toxicity, infection, inflammation, and preventing secondary lung injury has been shown to reduce morbidity and mortality in preterm infants [3]. CLD-reducing interventions start in the delivery room with timely delivery of exogenous surfactant and extend to the Neonatal Intensive Care Unit (NICU) where strategies are required for gentle ventilation, timely extubation, effective use of non-invasive ventilation, and optimized nutrition and growth. Standardized, evidence-based approaches are needed in NICUs to prevent secondary lung injury and reduce BPD/CLD. Herein, we describe a three-year quality improvement initiative to reduce CLD in the Golisano Children’s Hospital NICU in Rochester, NY, USA.

Our global aim was to reduce CLD, defined by the Vermont Oxford Network (VON) as the need for mechanical ventilation and/or supplemental oxygen at 36 weeks corrected gestational age (CGA), to the VON average (24%) for comparable level ≥ 3 NICUs. We assembled a multidisciplinary team of physicians, advanced practice providers, nurses, respiratory therapists, dieticians, pharmacists, information analysts, and nurse educators and formalized a large workgroup to identify specific interventions that would improve outcomes. By executing several fundamental elements of QI improvement: plan-do-study-act (PDSA) cycles, process measures, balancing measures, and data/outcome tracking, we were able to reach our goal and achieve sustained improvement.

## 2. Methods

### 2.1. Ethical Approval

This Quality Improvement project was reviewed by the University of Rochester Research Subjects Review Board (RSRB) office who determined that the proposed activity is not research involving human subjects as defined by Department of Health and Human Services (DHHS) and Food and Drug Administration (FDA) regulations. Therefore, patient consent, RSRB review, and approval was not required.

### 2.2. Context

The Golisano Children’s Hospital NICU is a 68-bed level 4 regional perinatal center in Rochester, NY, USA. The unit has a single patient room design, and approximately 1100 annual admissions of which approximately 170 are entered into the very low birth weight (VLBW) database, and who are the focus of our improvement work. As a regional perinatal center, we provide the highest level of care of critically ill newborns in the New York State Finger Lakes region, including those requiring cardiac and other advanced surgical services.

### 2.3. Interventions

Our multidisciplinary team reviewed baseline data about delivery room practices, initial ventilation strategies, use of high frequency ventilation, and medication/nutrition protocols, thus generating a key driver diagram to delineate a shared mental model for our approach (Figure 1). We identified several potential deviations from best practice including decreasing early surfactant administration, delays placing central lines and obtaining initial blood gases, and inconsistent ventilator initiation and weaning strategies in our unit over the years spanning 2011 to 2017. Notably, bubble continuous positive airway pressure (CPAP) was a new modality introduced to our unit in late 2017 before this quality improvement project began. One stated goal was taking a gestational age (GA) based approach so trends for ventilator use were established. As a result, three separate GA subgroups were identified to serve as the basis for proposed interventions: (1) ≤25 weeks (2) 26–27 weeks (3) 28–29 weeks. Furthermore, review of ventilation strategies during the first few postnatal weeks revealed trends for earlier initiation of high-frequency ventilation and more high frequency ventilation days, particularly among babies ≤25 weeks GA. The high variability and absence of unit-based guidelines suggested these changes in early ventilation strategies were a result of practice drift, allowing them to become a targeted focus of the first workgroups.

After initial data review, smaller workgroups formed with a directive to find potentially better practices based on GA subgroupings. One group concentrated on delivery room (DR) and immediate post-admission care, locally known as the “Golden Hour” workgroup while another focused on later preventative strategies (standard extubation criteria, non-invasive ventilation guidelines, gentle ventilation, high-frequency ventilation). Workgroups were multidisciplinary and comprised of a neonatologist, neonatal fellow or advanced practice provider, respiratory therapist, and/or nurse, to facilitate broader participation by all team members. Strategies for improvement included frequent chart auditing and feedback, use of plan–do–study–act (PDSA) cycles, and standardization of practices via several written guidelines (Table A1).

### 2.4. Project Year 2018

At baseline, significant variability in ventilation strategies existed in our unit, including variation in initial conventional mechanical ventilation settings and use and duration of high frequency ventilation. We chose to focus on standardizing practices whenever possible to reduce variability. Specifically, we observed significant variability in initial ventilator settings for VLBW infants requiring mechanical ventilation upon admission, which in turn impacted the time to extubation. As a result, we standardized our initial ventilator settings and encouraged timely extubation in March 2018. To identify potentially better practices, we looked to a recently published meta-analyses indicating that early surfactant administration and timely extubation reduces the risk of BPD, death, and intraventricular hemorrhage using INSURE or less invasive surfactant administration (LISA) strategies [4]. We then developed the GA based “Golden Hour” protocol (Figure 2) designed to standardize DR practices, promote early surfactant delivery, encourage timely extubation, and improve admission workflow. We launched the “Golden Hour” initiative in July of 2018 and also introduced an admission huddle using a standardized handoff from the delivery room to admitting providers to improve efficiency and reduce delays.

### 2.5. Project Year 2019

In 2019, our team focused on early respiratory management beyond the golden hour. Beginning in May 2019 we changed our exogenous surfactant product (to poractant alfa) to prepare for LISA and to reduce repeat surfactant dosing. In March 2019, our team developed non-invasive ventilation guidelines that promoted routine use of bubble CPAP as a first-line therapy. Ongoing data review suggested favorable successful extubation rates for infants ≥ 26 weeks gestational age. However, infants ≤ 25 weeks were exhibiting high rates of extubation failure. As a result, we developed non-invasive ventilation guidelines designed to standardize post-extubation CPAP (Bubble CPAP) and discourage other non-invasive modes (RAM CPAP or High Flow Nasal Cannula) in the immediate post-extubation period. We also recognized that increased emphasis on maintaining Bubble CPAP meant longer CPAP duration and observed increasing rates of nasal mucosa injury and breakdown. In response, we included education of front-line staff on ways to reduce pressure injury from CPAP, which included initiating a team of respiratory therapists, nurse educators, and bedside nurses to perform weekly bedside audits/rounds and reinforce strategies to reduce pressure injuries such as standard intervals for mask/prongs alternation and use of foam padding around pressure points.

### 2.6. Project Year 2020 and Ongoing Initiatives

Our unit adopted Bubble CPAP in November 2017 before this QI initiative was formalized. This was a change in the standard mode and CPAP interface, and we found higher rates of nasal breakdown in the early project years that was addressed with regular auditing. In early 2020, we formed a “Bubble CPAP Relaunch” focused on nursing and respiratory therapist education to gain additional expertise for troubleshooting issues with the seal and interface. Online education modules, a rolling cart in-service, display of posters, and bedside kits/cards were developed for this initiative. Pilot use of a Bubble CPAP alarm is underway to provide feedback and deliver more consistent pressures.

Review of our data suggested that a significant proportion of infants < 28 weeks GA required reintubation following extubation to bubble CPAP. Therefore, our team modified our non-invasive ventilation guidelines to promote use of Non-invasive positive pressure ventilation (NIPPV) as the primary extubation respiratory mode for infants < 28 GA, with the goal to transition to bubble CPAP by 7 days post-extubation. These guidelines also recommend extending Bubble CPAP use to 32 weeks CGA or 7 days post-extubation, whichever is later based on evidence suggesting improved lung growth and function with more sustained use [5,6,7].

The decreased administration volumes of our chosen surfactant preparation allowed our team to revisit and implement LISA. LISA offers several advantages: (1) the infant’s breathing mechanics help distribute the surfactant while on CPAP instead of with positive pressure ventilation and (2) an endotracheal tube is never placed, thus decreasing the need for unnecessary mechanical ventilation. After benchmarking with other centers utilizing LISA and obtaining the necessary catheters, a neonatal fellow in-serviced the NICU providers, respiratory therapists, and nurses. LISA launched in November 2020 with babies ≥29 weeks GA who were admitted to the NICU with respiratory distress syndrome on CPAP and required surfactant, in order to gain experience with the procedure in a familiar non-urgent setting, allowing every case to become its own PDSA cycle. Just-in-time training takes place before each LISA procedure for providers and nurses on the unit. Utilization of a video laryngoscope allows learners to better identify landmarks while manipulating the LISA catheter, thus offering guidance by providers with more experience. LISA criteria were recently broadened for repeat surfactant doses (in lieu of INSURE) and plans are underway to move LISA to the delivery room.

### 2.7. Measures and Analysis

The primary outcome measure was the VON CLD rate, defined as needing supplemental oxygen and/or mechanical ventilation at 36 weeks CGA among inborn infants ≤ 29 6/7 weeks gestation or birth weight < 1500 g, excluding DR or deaths occurring within the first 12 h of life. Infants were also classified as CLD if discharged/transferred before 36 weeks CGA on oxygen. Secondary outcome measures included failed extubation rate and need for postnatal steroids, repeat surfactant, or discharge diuretics. Process measures (Table A2) included compliance with DR surfactant guidelines and initial ventilator settings. Our balancing measures (Table A3) were failed extubation rate and use of postnatal steroids as described above. All data were gathered from a local database and/or extracted from the electronic medical record. Statistical Process control charts (QI Macros, KnowWare International, Inc., Denver, CO, USA) were used to display and analyze data over time. Special cause variation was determined using suggested criteria specific to health care [8]. Pre- and post- “Golden Hour” CLD rates were compared by chi-squared analysis.

## 3. Results

Patient demographics were similar before and after the start of our QI initiative with the exception of decreased maternal chorioamnionitis diagnoses after the project started (Table 1). Our baseline rate of CLD, using data from January 2016 through December 2017, was 33.5%. Special cause variation was noted (rule of shift) following implementation of standardized practices with the Golden Hour protocol in July 2018, and the rate of CLD decreased by 51% to a new baseline rate of 16.5%. (Figure 3). To feel confident in our change, we waited 10 months before calculating a new center line. GA Subgroup analyses showed decreases in BPD rates comparing before and after the QI initiative started among all gestational age groups, most evident in the 26–27 weeks GA infants (Figure 4). The percent of infants born ≤ 28 weeks GA receiving prophylactic surfactant in the DR also increased following the Golden Hour protocol from a baseline average of 80.8% to 98.1% (Figure 5), strengthening our conclusion that standardization of practices had a significant impact on our rates of CLD.

Since gentle ventilation strategies that avoid prolong mechanical ventilation protect against lung injury, an important component of our baseline data review was tracking intubation rates over time. A local database was queried for the presence of an endotracheal tube each day through the first 56 postnatal days (Figure 6, black lines). Specifically, the vast majority (>90%) of infants ≤ 25 weeks gestational age needed ventilation for ≥2 weeks with approximately 50% still intubated at 6 weeks of age. Early extubation was more successful (approximately 50%) before day of life (DOL) 3 with more gradual improvement among infants 26–27 weeks GA. Finally, most (>90%) of infants 28–29 weeks GA were managed noninvasively in the first postnatal week, or never required intubation for the entire hospitalization.

Intubation rates among all GA groups improved after starting the “Golden Hour” initiative (Figure 6, gray lines). Some infants ≤ 25 weeks GA were extubated by DOL 3, but these extubations were rarely successful, as the intubation rate climbed back above 85% during the second postnatal week when many infants experience an increase in ventilator and oxygen requirements, termed “pulmonary deterioration” [9]. However, a sustained decrease in intubation rates occurred starting at approximately DOL 28 through DOL 56. With focus on standardizing extubation readiness and mode (non-synchronized non-invasive PPV), intubation rates for 26–27 weeks infants reduced by approximately 50%. Finally, modest improvements in intubation rates were observed in 28–29 weeks infants though baseline intubation rates were low in these babies.

Despite DR surfactant administration and encouraging early extubation, an important balancing and cost-savings measure required tracking repeat surfactant dosing, which occurs in our NICU for sustained FIO2 requirement >30% before DOL 3. We did not observe an increase in repeat surfactant doses after the Golden Hour Bundle test of change but observed special cause variation shortly after changing surfactant preparations (Figure A1) that was most pronounced in babies of greater gestational age. Taken together, the concentration on non-invasive modes of ventilation allowed providers to be more comfortable with early extubation, and decreased extubation failure rates among all GA subgroups.

We hypothesized that with improving CLD rates, the use of postnatal glucocorticoids and/or discharge diuretics would also decrease. Conversely, increasing use of steroids and diuretics may reflect providers adding therapies because infants were slower to wean approaching discharge. The use of postnatal glucocorticoids for respiratory insufficiency has not increased and remains below 10%. Discharge diuretic use decreased from 44 to 22% (Figure A2) after observing special cause variation in December 2018, approximately 5 months after the “Golden Hour” bundle was initiated. For the first time, there were consecutive months where zero VON CLD eligible infants were discharged on diuretics, which decreases the burden for parental follow-up and pediatric pulmonologists who manage these medications outpatient.

## 4. Discussion

### Summary and Interpretation

We have shown that through a comprehensive multidisciplinary QI initiative sustained improvement in CLD rates can be achieved in a level 4 NICU. Our team applied core QI principles that included establishing an aim statement and key driver diagram, testing changes with iterative PDSA cycles, and measuring primary and secondary outcomes over time. This robust approach allowed us to improve CLD rates without increasing harm or other balancing measures that might increase risk for other prematurity-related morbidities (i.e., reintubation and intraventricular hemorrhage). We have successfully changed our unit culture and work is ongoing to with a revised SMART aim goal to reduce VON CLD below the first quartile for comparable NICUs (17%).

The current observed monthly BPD rates have decreased by approximately 51%, and improvements have been sustained for >18 months. Several tests of change correlated with special cause variation, the biggest of which was adoption of the “Golden Hour” protocol. Stratifying our protocols by gestational age where applicable was a key component of our findings. This process started with a detailed interrogation of our baseline data by gestational age to see where ventilator strategies clustered together. For example, we ascertained that infants ≤ 25 weeks would require prolonged ventilation and thus have focused our efforts on gentle ventilation strategies and non-invasive ventilation modes that reduce extubation failure. Conversely, infants 26–27 weeks are more likely to be successfully managed with non-invasive modes after one dose of surfactant, thus avoiding the consequences of mechanical ventilation and reducing lung injury. There were several reasons for taking a GA-based approach: (1) stratifying DR practices in this manner (instead of birth weight) allows the team to prepare in advance because birth weight is not known before delivery (2) practice changes can be targeted to specific gestational age ranges (i.e., automatic delivery room surfactant for infants ≤ 27 weeks) without unnecessary interventions or cost to other groups (3) additional resources may be needed for infants at the extremes of viability and (4) primary (CLD) and secondary (caffeine, diuretics, postnatal steroids, extubation success) outcomes can be similarly stratified to better understand barriers to change and identify additional areas for targeted PDSA cycles. Beyond GA subgroups, efforts are underway to individualize extubation success probabilities utilizing the electronic medical record, which can aid the medical team in real-time, thus reinforcing practice changes.

Sustained improvement or “holding the gains” can be challenging in complex health care systems but are made possible through several factors. First, the use of frequent audits for process measure compliance such as monitoring DR surfactant compliance allows to determine when deviations occur. Recently, one such cause was identified which prompted a chart review of the DR course of each infant, identifying contributory systems issues and offering opportunities for reeducation. There are further opportunities to use the electronic medical record to sample ventilation data with increasing frequency and develop machine learning algorithms that can project a neonate’s respiratory course. Finally, introducing other ventilation modes such as synchronized non-invasive positive ventilation was on the horizon for our unit, until the COVID-19 pandemic stretched ventilator resources and delayed its implementation.

The limitations to our initiative include that it reflects practice changes occurring in a single NICU, some of which may not be broadly generalizable to other units. Personnel availability (i.e., DR support for more involved procedures such as LISA and the NICU footprint itself (i.e., single patient rooms, proximity to delivery room) can limit broader applicability of our local practices. Additionally, a key component to any quality improvement or practice change is educating providers and staff. We are fortunate to have dedicated nurse educators to efficiently disseminate new information and data analysts to track the broad range of outcomes necessary for the success of this initiative. In addition, the population of VON CLD eligible infants skews heavily toward the older gestational ages (>28 weeks), making sustained gains difficult since CLD rates are relatively low. Finally, there were other non-respiratory management changes such as standard management of the patent ductus arteriosus and ongoing neonatal clinical trials that make it difficult to “unbundle” practice changes from other concurrent work in the NICU.

As often occurs with improvement work, we believe our learned experience from older infants can “trickle down” to lower gestational age groups as these infants are most vulnerable to lung injury. Many infants can be managed non-invasively with best practices, and those that require mechanical ventilation require careful attention to minimize ventilator-associated lung injury. To supplement our overall improvements in CLD rates, there are still ongoing efforts in our NICU targeted toward our smallest infants. A separate “Small Baby Program” quality improvement initiative is underway which will incorporate respiratory management guidelines with best practices in other disciplines (nutrition, skin care, etc.). Reducing other negative sequalae of prematurity (e.g., NEC, late onset infection, and intraventricular hemorrhage), may further potentiate our improved respiratory outcomes, thus increasing morbidity-free survival, and allowing our neonates to achieve the best possible outcome.

## Figures and Tables

**Figure 1 children-08-00301-f001:**
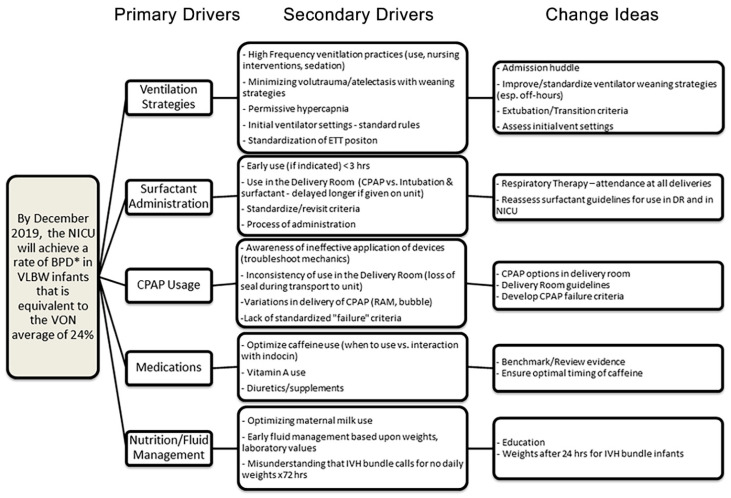
Key driver diagram for reducing Chronic Lung Disease linking change ideas to primary and secondary drivers. Abbreviations: NICU—Neonatal Intensive Care Unit; BPD*—need for supplemental oxygen at 36 weeks corrected gestational age; VON—Vermont Oxford Network; CPAP—continuous positive airway pressure, ETT—endotracheal tube; hrs—hours.

**Figure 2 children-08-00301-f002:**
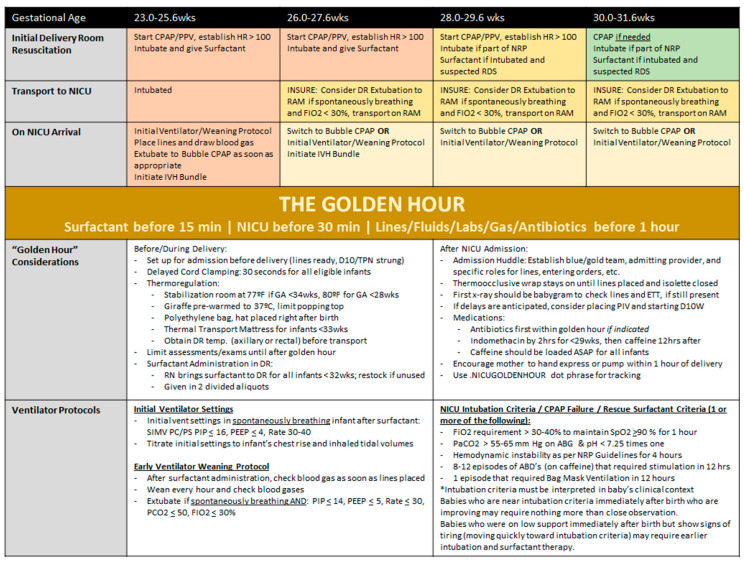
The “Golden Hour” gestational-age based protocol to standardize DR and admission practices. Abbreviations: PPV—positive pressure ventilation; HR—heart rate; NRP—Neonatal Resuscitation Program; DR—delivery room; OR—operating room; TPN—total parenteral nutrition; ABG—arterial blood gas; SIMV—synchronized intermittent mandatory ventilation; PC/PS—pressure control pressure support; PEEP—positive end expiratory pressure; PCO2/PaCO2—partial pressure of carbon dioxide; SpO2—oxygen saturation; ABD—apnea, bradycardia, and desaturation; wks—weeks; hrs—hours.

**Figure 3 children-08-00301-f003:**
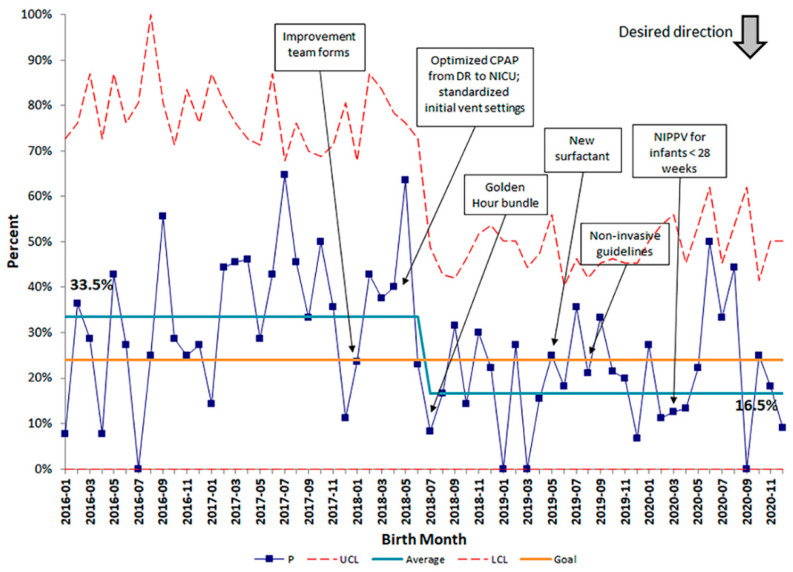
Annotated SPC p-chart displaying the percent of infants ≤ 29 weeks GA or birth weight < 1500 g with Chronic Lung Disease, defined as need for supplemental oxygen and/or mechanical ventilation at 36 weeks gestational age. Special cause variation occurred at one time point. Orange line represents the SMART aim goal rate of 24%. Red dashed lines represent the upper control limit (UCL) and lower control limit (LCL). *N* = 711 total admissions.

**Figure 4 children-08-00301-f004:**
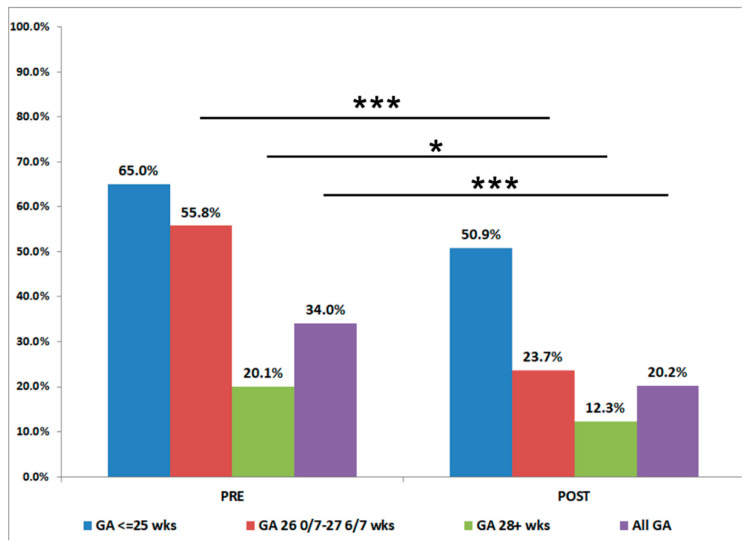
Comparison of Chronic Lung Disease (CLD) rates before (PRE) and after (POST) the QI initiative. The overall (purple) rate of CLD decreased from 34.0 to 20.2%. Gestational Age Specific CLD rates decreased among infants 28–29 weeks (green) and 26–27 weeks (red), and <25 weeks (blue). * *p* < 0.05, *** *p* < 0.001. Abbreviations: SPC—statistical process control; SMART—Specific, measurable, achievable, relevant, time-bound.

**Figure 5 children-08-00301-f005:**
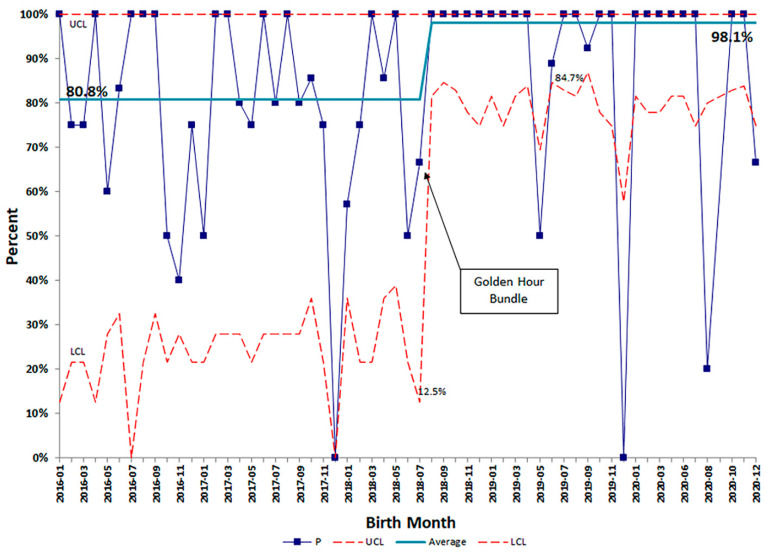
SPC p-chart for delivery room surfactant administration. Special cause variation was observed after initiation of the golden hour bundle, raising the control line from 80.8% to 98.1%. Subsequent months where variation occurred outside of the lower control limit prompted chart review and reeducation of providers and staff. Abbreviations: SPC—statistical process control; P—proportion; UCL—upper control limit (upper dotted red line); LCL—lower con-trol limit (lower dotted red line).

**Figure 6 children-08-00301-f006:**
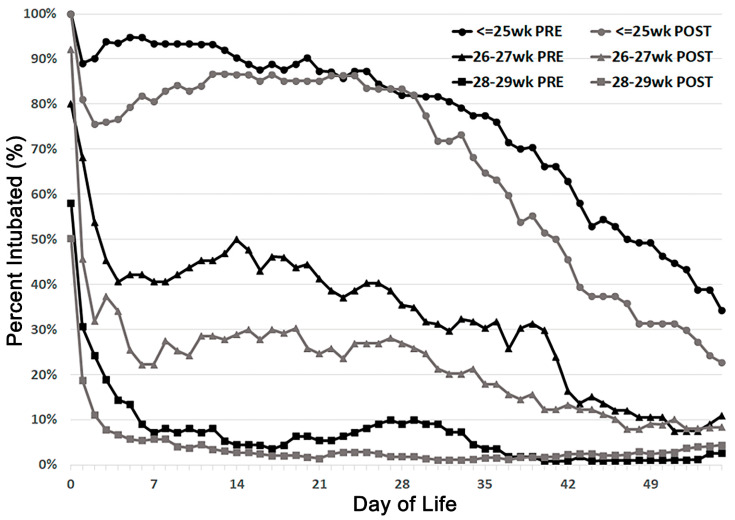
Intubated babies in two epochs: Pre-(January 2016 to July 2018, black solid line) and post-golden hour (July 2018 to December 2020, gray solid line. ≤25 weeks circles; 26–27 weeks triangles; 28–29 weeks squares) show decreases in intubation rates in all gestational age groups throughout the study period.

**Table 1 children-08-00301-t001:** Population Demographics. Patient characteristics were similar before and after the intervention period with the exception of decreased diagnoses of maternal chorioamnionitis.

	Before Intervention (1 January 2016–31 March 2018)	After Intervention (1 April 2018–31 December 2020)	*p*-Value
Gestational Age < 30 weeks	205 (63.7%)	305 (65.5%)	0.61
Birth Weight < 1500 g and GA < 29 weeks	117 (36.3%)	161 (34.5%)	0.61
Male Sex	172 (53.4%)	223 (47.9%)	0.12
Inborn	292 (90.7%)	420 (90.1%)	0.80
Intrauterine Growth Restriction	19 (5.9%)	28 (6%)	0.94
Maternal Pre-eclampsia	124 (38.5%)	199 (42.7%)	0.23
Maternal Chorioamnionitis	101 (31.4%)	115 (24.7%)	0.03
Antenatal Steroids	300 (93.2%)	429 (92.1%)	0.56
Postnatal Steroids	43 (13.4%)	60 (12.9%)	0.84

Abbreviations: GA—gestational age.

## Data Availability

Not applicable for Quality Improvement work.

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
