# Peer review of "A Quality Improvement Initiative to Reduce Bronchopulmonary Dysplasia in a Level 4 NICU—Golden Hour Management of Respiratory Distress Syndrome in Preterm Newborns"

_children, 2021, doi:10.3390/children8040301_

Round 1

Reviewer 1 Report

A Quality Improvement Initiative to Reduce Bronchopulmonary Dysplasia in a Level 4 NICU - Golden Hour Management of Respiratory Distress Syndrome in Preterm Newborns  

By Andrew M. Dylag, et al.

This paper shows a “Quality Improvement” approach to reduce the BPD incidence, particularly in the 26-27 wk GA, in preterm newborns affected by RDS. The Authors elicit the main steps and the changes in their therapeutic skills and care of admitted newborns, and the golden hour in their decisions making. The Authors rightly showed useful protocols from DR onward, to reach the proposed aim, i.e. the decrease in incidence of BPD. This requires many good/standardized practices and frequent neonatologists’ discussions of the deviations/problems around these innovative approaches. The paper is clear in its home message and also in the reproducibility for implementing the program in other tertiary NICUs. Figures could be reduced in size to improve their comprehension. It is fascinating that similarly to new therapies for BPD, i.e. drugs or cells therapies, a “Quality Improvement” policy Small Baby Program in respiratory care, could exert the fundamental objective of decreasing BPD.

Author Response

Reviewer Comment: Figures could be reduced in size to improve their comprehension. It is fascinating that similarly to new therapies for BPD, i.e. drugs or cells therapies, a “Quality Improvement” policy Small Baby Program in respiratory care, could exert the fundamental objective of decreasing BPD.

Authors Response: We appreciate your response.  We included this volume of figures to highlight how our QI methodology integrates with more traditional outcome tracking, and included or process and flow charts so other may utilize these same techniques.  If the editors believe less figures are warranted, we would certainly be open to moving some to a supplement.

Reviewer 2 Report

I read with interest the paper by Dylag and colleagues about a quality improvement strategy to prevent bronchopulmonary dysplasia.

Authors should be commended for their effort in practice amelioration and serving as an example for other clinicians.

My only major concern looking at the described results is the total absence of a description of population characteristics. Were the two populations before and after bundle application comparable? (same gestational age, birth weight, IUGR incidence, preeclampsia incidence, mother chorio, sex, prenatal steroids, inborn/outborn, postnatal steroids etc..)
To state that after the application of the new protocols surfactant was administered earlier, is a possible result independently from the population.
Saying that newborns were more frequently extubated at the same day of life is on the contrary a speculation if the population characteristics aren’t explicit. Moreover, affirming that BPD had a lower incidence without describing two comparable populations is not possible.

In summary the work is well written and structured, describes an interesting and positive approach toward amelioration, inspiring possible similar projects in other centers. The methods and protocol development are well described and results are encouraging.

Author Response

Reviewer Comment: My only major concern looking at the described results is the total absence of a description of population characteristics. Were the two populations before and after bundle application comparable? (same gestational age, birth weight, IUGR incidence, preeclampsia incidence, mother chorio, sex, prenatal steroids, inborn/outborn, postnatal steroids etc..)

Authors Response: We have included a Table 1 with a summary of patient demographics before and after our QI initiative.  Most demographics were similar with the exception of decreased chorioamnionitis.  This information could be part of the manuscript, part of supplemental data, or for review only.

Round 2

Reviewer 2 Report

No further requests.